# RGM: Reconstructing High-fidelity 3D Car Assets with Relightable 3D-GS Generative Model from a Single Image

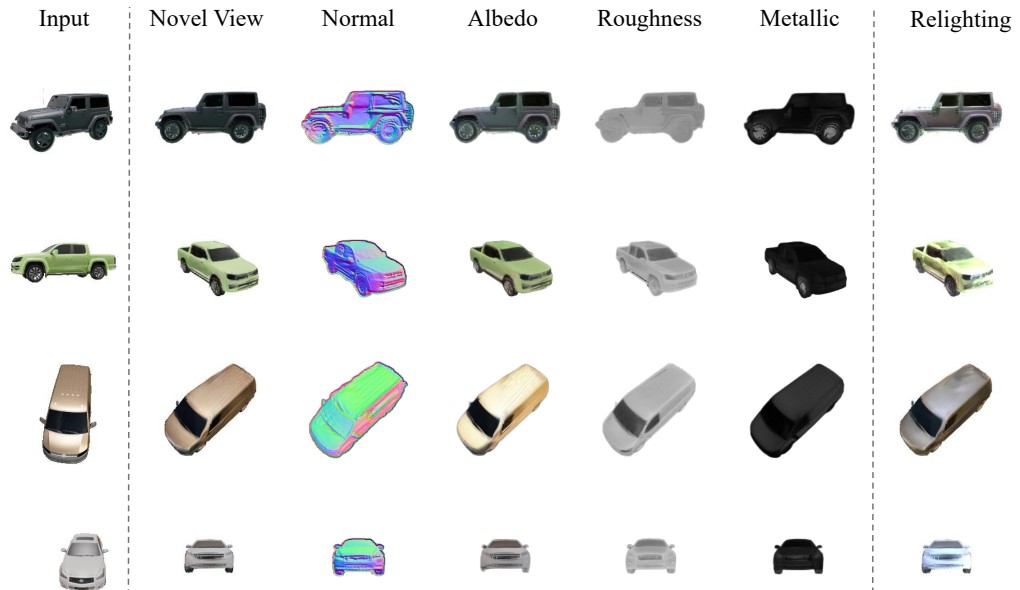

Figure 1: Our method generates high-quality 3D cars and corresponding material properties from the single image, which allows for realistic relighting under different lighting conditions.

## ABSTRACT

The generation of high-quality 3D car assets is essential for various applications, including video games, autonomous driving, and virtual reality. Current 3D generation methods utilizing NeRF or 3D-GS as representations for 3D objects, generate a Lambertian object under fixed lighting and lack separated modelings for material and global illumination. As a result, the generated assets are unsuitable for relighting under varying lighting conditions, limiting their applicability in downstream tasks. To address this challenge, we propose a novel relightable 3D object generative framework that automates the creation of 3D car assets, enabling the swift and accurate reconstruction of a vehicle's geometry, texture, and material properties from a single input image. Our approach begins with introducing a large-scale synthetic car dataset comprising over 1,000 high-precision 3D vehicle models. We represent 3D objects using global illumination and relightable 3D Gaussian primitives integrating with BRDF parameters. Building on this representation, we introduce a feed-forward model that takes images as input and outputs both relightable 3D Gaussians and global illumination parameters. Experimental results demonstrate that our method produces photorealistic 3D car assets that can be seamlessly integrated into road scenes with different illuminations, which offers substantial practical benefits for industrial applications.

## 1 INTRODUCTION

Generating high-quality 3D car assets plays a vital role in various applications like digital city, autonomous driving, and virtual reality, among others. Traditionally, the 3D modeling of vehicle assets is hand-crafted by skilled artists using Computer-Aided Design (CAD) software. The modeled cars are then integrated into rendering engines such as Blender Blender (2019) or Unreal Engine Epic Games for photorealistic rendering. However, establishing such a 3D asset can be a cumbersome task, especially when there is a need to model 3D assets based on images of real-world cars. In such cases, the modeling artists are required to manually design and adjust the shape, texture and material of cars in detail to match the reference image accurately. To alleviate the burden on artists and facilitate complex car modeling, in this work, we focus on generating 3D car assets automatically with a data-driven method. Specifically, given an input image of a random 3D car, we aim to swiftly and precisely reconstruct its 3D geometry, texture, and material like roughness and metallic. This allows applications like relighting and holds substantial promise for industrial design.

In recent years, there has been a revolution in 3D generation with the rise of diffusion models. Some works Poole et al. (2022); Lin et al. (2023); Ornek et al. (2023) integrate NeRF Milden-hall et al. (2021) or 3D GS Kerbl et al. (2023) with diffusion models to generate 3D objects using text prompts. However, these optimized-based methods typically face challenges like slow generation speed and unstable training processes. Other methods Shi et al. (2023); Liu et al. (2023a; 2024) extend 2D diffusion models to generate consistent multi-view images by introducing viewpoint awareness. For example, Wonder3dLong et al. (2024) generates consistent multi-view RGB images with corresponding normal maps via a cross-domain diffusion model, with the input of a reference image and camera parameters. However, existing multi-view image generation models only map a single image to multiple views, necessitating supplementary modules like NeuSWang et al. (2021) to reconstruct the 3D shape and texture. These object-specific models often demand a lot of training time, ranging from minutes to hours. Recently, several works Tang et al. (2024a); Xu et al. (2024); Tochilkin et al. (2024); Hong et al. (2023) have leveraged transformer architecture for a feed-forward 3D generation that takes images as input and outputs 3D representation like NeRF or 3D-GS. These methods offer rapid 3D model generation through fast feed-forward inference and provide more precise control over the generated outputs.

Despite these advancements, the 3D generation of vehicles still struggles with issues such as insufficient detail and inconsistent geometry. These limitations are primarily due to the lack of large-scale car datasets that could support the learning of generalizable priors for vehicles. Besides, current 3D generation methods primarily focus on the geometry and texture of objects, often neglecting material modeling. In these methods, objects' shading colors are typically baked into the surface, showing only the lighting effect under specific illumination. As a result, the generated objects are unsuitable for relighting with different illumination and require manual material assignment for downstream applications, thereby limiting their practicality for industrial 3D asset generation.

To address the aforementioned problems, we propose a large-scale synthetic car dataset and a novel relightable 3D object generation pipeline, offering efficient solutions for rapid and precise 3D car asset creation. First, we collect and construct a synthesized vehicle dataset named Carverse, which contains over 1,000 high-precision 3D vehicle models. These models are rendered with different materials and under different lighting to obtain multi-view images and camera poses. With the dataset, we introduce a novel relightable 3D-GS generative model (RGM) that: (1) We model a 3D scene with global illumination and a set of relightable 3D Gaussians, where each Gaussian primitive is associated with a normal direction and BRDF parameters including albedo, roughness, and metallic. (2) we adopt a U-Net Transformer as an image-conditioned model, which outputs both the relightable 3D-GS and global illumination parameters. With this pipeline, we successfully reconstruct the geometry, appearance, and material of a 3D car from an image, enabling the realistic insertion of the car into different scenes. We further present an autonomous driving scene simulation pipeline to demonstrate the practicality of RGM.

Fig. 1 provides qualitative results of the reconstructed cars with normal and material maps. Experiments demonstrate that our method outperforms previous approaches both quantitatively and qualitatively. Additionally, we showcase the effectiveness of relighting and integrating the generated car assets into road scenes. The results achieve photo-realistic performance, providing significant benefits for a wide range of AR/VR applications.

## 2 Related Works

### 2.1 Generative Model

The goal of generative model is to produce new samples with similar statistical characteristics to the training data by learning the underlying distribution. In recent years, there has been a revolution in generative model, with groundbreaking networks named diffusion models Sohl-Dickstein et al. (2015). Diffusion models have demonstrated significant success in image synthesis Rombach et al. (2022); Zhang et al. (2023); Ye et al. (2023); Podell et al. (2023) by generating new samples through the gradual denoising of a normally distributed variable. They have also inspired numerous works to generate 3D assets. Many works Poole et al. (2022); Lin et al. (2023); Tang et al. (2023); Ornek et al. (2023) integrate NeRF Mildenhall et al. (2021) or 3D Gaussian Kerbl et al. (2023) with diffusion models to generate 3D objects using text prompts. Additionally, several works Shi et al. (2023); Tang et al. (2024b); Liu et al. (2023a; 2024) extend 2D diffusion models to generate consistent multi-view images by introducing viewpoint awareness. Moreover, some works Richardson et al. (2023); Chen et al. (2023a); Youwang et al. (2023); Zeng (2023) focus on generating texture for 3D assets by extending text-conditional generative models. However, these works typically faces challenges like slow generation speed and unstable optimization process. Recently, servral works Tang et al. (2024a); Xu et al. (2024); Tochilkin et al. (2024); Hong et al. (2023) utilize transformer architectures for feed-forward processes, converting images into 3D representations such as NeRF or 3D-GS . These approaches enable rapid 3D model generation through efficient feed-forward inference. Nevertheless, these methods primarily concentrate on reconstructing shape and texture, which restricts their utility in practical industrial applications. In this study, we investigate a fast 3D generation pipeline capable of producing high-quality 3D car assets as well as material properties.

### 2.2 3D Gaussian Splatting

3D Gaussian splatting (3DGS) employs 3D Gaussian primitives to represent the 3D scene and alpha blending to render the image. In comparison to the NeRF, the 3DGS method exhibits superior rendering quality and speed. A number of previous studies have addressed the misrendering issues caused by viewpoint changes during 3DGS rendering. Mip-splattingYu et al. (2024) and SA-GSSong et al. (2024) address the frequency inconsistency of Gaussian primitives when the camera is zoom-in or zoom-out by employing different 3D or 2D filters. Other works enhance the rendering quality by modifying the form of Gaussian primitives. For example, 2DGSHuang et al. (2024) compresses Gaussian primitives into 2D facets, enhancing the Gaussian geometry and facilitating superior normal vector rendering. 3d-HGSLi et al. (2024) disassembles the Gaussian into a half-Gaussian, enhancing its scene representation. Furthermore, certain worksLu et al. (2024); Ren et al. (2024); Yan et al. (2024b); Yang et al. (2024) endeavor to restrict the use of GS to achieve more compact scene structures. Besides, some works have employed various deep learning modules (e.g., GS in the DarkYe et al. (2024b), Deblur-GSChen & Liu (2024)) to address the issue of noisy images. Others have utilized GS to model large-scale streetscapes (e.g., Street GSYan et al. (2024a), AutosplatKhan et al. (2024)). Moreover, some works employ the modeling of physical properties in 3DGS through the representation of BRDF functions, as exemplified by Relightable GSGao et al. (2023) and 3D Gaussian Splatting with Deferred ReflectionYe et al. (2024a). In this study, we also use 3DGS to model a 3D car and integrate it with physical attributes to model the vehicle under arbitrary lighting conditions and to enable light editing of the vehicle.

### 2.3 Vehicle Modelling and Datasets

To facilitate the development of autonomous driving and vehicle modeling, researchers have created a series of datasets. Real-world driving datasets like KITTIGeiger et al. (2013), Waymo, and nuScenesCaesar et al. (2020), offer comprehensive sensor data that are essential for capturing realistic driving scenarios. These datasets focus on tasks such as object detection, tracking, and trajectory prediction. However, their sparse viewpoints make them inadequate for vehicle reconstruction. Simulation datasets from platforms like CARLADosovitskiy et al. (2017) allow researchers to create diverse and controlled driving environments; nevertheless, the vehicles in these datasets tend to be of low quality, rendering them unsuitable for realistic downstream applications. SRN-CarChang et al. (2015) and ObjaverseDeitke et al. (2023) compile 3D car models from various repositories and online resources, but their quality is insufficient to meet the demands of real vehicles for au-

| Dataset | Instances | Views | Quality | Diversity | Normal | Material |
|---------|-----------|-------|---------|-----------|--------|----------|
| **SRN-Car** Chang et al. (2015) | 2,151 | 250 | Low | Medium | ✗ | ✗ |
| **MVMC** Zhang et al. (2021) | 576 | ∼10 | High | Low | ✗ | ✗ |
| **Objvaverse** Deitke et al. (2023) | 511 | - | Medium | Low | ✗ | ✗ |
| **CarStudio** Liu et al. (2023b) | 206,403 | 1 | High | High | ✗ | ✗ |
| **3DRealCar** Jiang et al. (2024) | 2,500 | ∼200 | High | High | ✗ | ✗ |
| **Carverse** | $1,006 \times 5$ | 100 | High | High | ✔ | ✔ |

Table 1: Statistic comparison with concurrent vehicle datasets.

tonomous driving. MVMCZhang et al. (2021), collected from car advertisement websites, is too sparse to generate high-quality 3D car models. The dataset CarStudioLiu et al. (2023b) provides images from 206,403 car instances, while the 3DRealCar datasetSong et al. (2019) includes multi-view images of 2,500 car instances derived from real vehicle data. However, users do not have access to additional attribute information, such as normals, metallic properties, and roughness. In this paper, we introduce a novel vehicle dataset named Carverse, which not only presents high-quality images from a diverse set of cars but also includes geometry and material information, facilitating a deeper understanding of vehicles. Tab. 1 shows statistic comparisons with previous datasets.

## 3 METHOD

In this section, we present our relightable car assets generation framework, shown in Fig. 3, which reconstructs a car's geometry, materials, and illumination from images under unknown lighting. We first introduce a novel large-scale synthetic car dataset (Sec. 3.1). Then, we present relightable Gaussian primitives to represent 3D objects with material properties (Sec. 3.2) and propose a relightable 3D-GS generative model that infers from images and output the relightable Gaussians (Sec. 3.3). Finally, we show an autonomous driving simulation application in Sec. 3.4 to integrate the generated car into road scenes and render photorealistic images.

### 3.1 CARVERSE DATASET CONSTRUCTION

As a crucial component of autonomous driving simulation, the manual creation of vehicle models by artists is an exceptionally time-consuming process. In this paper, we present a data-driven pipeline for automatic vehicle generation, which necessitates a large-scale, high-quality vehicle dataset. Existing datasets are often limited by low quality and a restricted variety of vehicles. To overcome these challenges, we introduce **Carverse**, a high-quality synthetic dataset designed to support research in vehicle reconstruction and related tasks. Carverse is constructed using Blender's physics engine Blender (2019) and the ray-tracing Blender Cycles rendering engine, enabling the generation of photorealistic images with diverse materials and global illumination settings.

To develop a comprehensive and realistic dataset, we sourced over 1,000 high-quality 3D vehicle models and 3,000 distinct HDR maps from online repositories. The dataset includes a wide range of vehicle types, such as sedans, sports cars, vans, and trucks. The 3D vehicle models were divided into training and testing sets with 1,006 as the train set and 50 as the test set. To ensure dataset diversity, each model was subjected to random z-axis rotations and minor flipping along the XY plane prior to rendering. Additionally, HDR maps, vehicle colors, and materials were randomly varied to simulate different illumination and surface properties. Each vehicle in the training set was rendered 5 times with randomly assigned textures and lighting conditions, yielding a total of 5,030 samples across the 1,006 vehicles. For each sample, we captured ground truth data from 100 distinct camera poses, including monocular RGB renderings, camera pose information, normal maps, and material maps such as albedo, roughness, and metallic. In the test set, 50 vehicle samples were rendered, with one image used as input and 30 random views for benchmarking, along with a mesh file for geometry evaluation. Fig. 2 illustrates the dataset construction pipeline and showcases sample diversity.

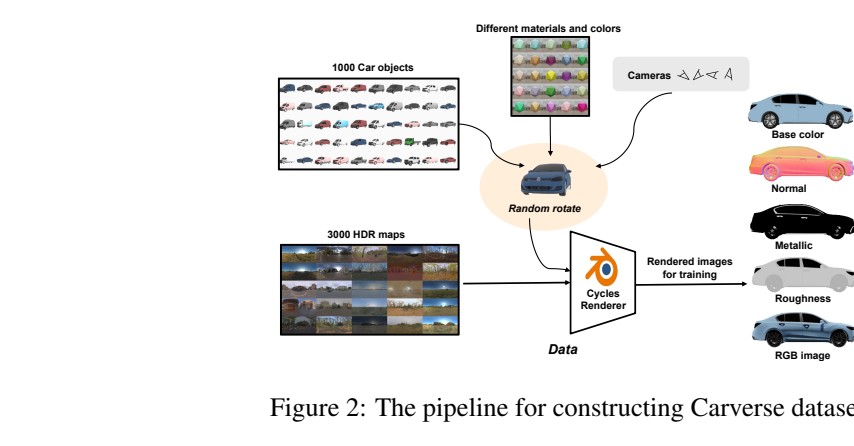

Figure 2: The pipeline for constructing Carverse dataset.

## 3.2 RELIGHTABLE 3D GAUSSIAN

We aim to efficiently generate 3D objects with material properties from a single image, allowing seamless application to downstream tasks. Recent advancements in 3D object generation have leveraged transformer architectures for rapid feed-forward 3D generation, processing images as inputs and producing 3D representations such as 3D-GS. Specifically, in the representation of 3D-GS, each 3D Gaussian primitive $G_i(x)$ is mathematically defined as:

$$G_i(x) = e^{-\frac{1}{2}(x-\mu_i)^T \Sigma_i^{-1}(x-\mu_i)}, \tag{1}$$

$x \in \mathbb{R}^3$ is a random 3D coordinate, $\mu_i \in \mathbb{R}^3$ stands for the mean vector of the Gaussian, and $\Sigma_i \in \mathbb{R}^{3\times 3}$ is the covariance matrix. To render the Gaussians into the image space, each pixel $p$ is colored by $\alpha$-blending $K$ sorted Gaussians overlapping $p$ as:

$$C(p) = \sum_{i=1}^{K} c_i \alpha_i \prod_{j=1}^{i-1} (1 - \alpha_j), \tag{2}$$

where $\alpha_j$ is derived by projecting the 3D Gaussian $G_i$ onto pixel $p$ and multiplying by its opacity, and $c_i$ denotes the color of $G_i$. Using a differentiable tile-based rasterizer Kerbl et al. (2023), all Gaussian attributes can be optimized in an end-to-end manner.

However, limited by the representation of 3D-GS, existing feed-forward generative models primarily focus on geometry and texture, overlooking material properties and lighting effects, which typically require manual adjustment before the generated objects can be applied to downstream applications. To address this, we extend the representation of 3D-GS by modeling a 3D object with global illumination and a set of relightable 3D Gaussians, where each Gaussian primitive is associated with a normal direction and BRDF parameters. This enriched representation facilitates the generating of material properties and supports object editing tasks such as relighting.

To model the shading color of objects under different lighting conditions, we use the physically-based rendering equation to compute the outgoing radiance $L_o$ at an intersected surface point $x$:

$$L_o(x, \omega_o) = \int_{\Omega^+} f_r(x, \omega_i, \omega_o) L_i(x, \omega_i) \cos\theta d\omega_i, \tag{3}$$

where $f_r$ is the BRDF function, $\Omega^+$ is the positive hemisphere determined by the surface normal $n$ at point $x$, $\omega_o$ is the direction of the outgoing ray, $\omega_i$ represents all possible incident ray directions in $\Omega^+$, $L_i$ is the incident radiance from direction $\omega_i$, and $\theta$ is the angle between $\omega_i$ and $n$. By integrating Eq. 3 with 3D-GS, we present the representation of relightable Gaussian primitives as follows.

**Normal modelling**. Surface normals are critical in rendering equations to determine the shading of a surface point. Though we can compute the pseudo normal $\widetilde{N}$ with the rendered depth map $D$ of 3D-GS, it is not differentiable and is usually imprecise. To accurately model the geometry of 3D objects, we first enhance 3D-GS representation by integrating it with a learnable normal direction. Specifically, for each 3D Gaussian primitive $G_i$, we introduce an additional 3D unit vector $n_i \in \mathbb{S}^2$.

Then we can obtain a rendered normal map N through $\alpha$-blending as:

$$\mathrm{N(p)} = \sum_{\mathrm{i}=1}^{\mathrm{K}} \mathrm{n_i} \alpha_\mathrm{i} \prod_{\mathrm{j}=1}^{\mathrm{i}-1} \left(1 - \alpha_\mathrm{j}\right). \tag{4}$$

We use the pseudo normal map $\widetilde{\mathrm{N}}$ as supervision to guide learning of the normal attributes, with a consistency loss formulated as:

$$\mathcal{L}_{\mathrm{consistency}} = \|\mathrm{N} - \widetilde{\mathrm{N}}\|_2^2. \tag{5}$$

**Material modelling**. We employ a simplified Disney BRDF function $f_r$ Burley & Studios (2012) to capture the effects of spatially varying materials on light reflection. The BRDF describes the relationship between incoming and outgoing light at a surface point, illustrating how material properties influence the reflection. Specifically, in the BRDF formulation $\mathrm{f_r(x, \omega_i, \omega_o; n, b, r, m)}$, the variables are defined as follows: n represent the surface point's normal, while $\mathrm{b} \in [0, 1]^3$, $\mathrm{r} \in [0, 1]$, and $\mathrm{m} \in [0, 1]$ represent diffuse albedo, and roughness and metallic parameter respectively. For each Gaussian primitive $\mathrm{G_i(x)}$, we associate these parameters to model the material properties, therefore the variable for a relightable Gaussian includes $\mathrm{G_i} = \{\mu_\mathrm{i}, \Sigma_\mathrm{i}, \mathrm{c_i}, \alpha_\mathrm{i}, \mathrm{n_i}, \mathrm{b_i}, \mathrm{r_i}, \mathrm{m_i}\}$. These attributes can be rendered into corresponding attribute maps (e.g., albedo map B, roughness map R and metallic map M) using the differentiable tile-based rasterizer. Consequently, they can be optimized through a $\mathrm{L_2}$ loss $\mathrm{L_{material}}$ with the ground truth maps.

**Illumination modelling**. To enable the insertion of objects into scenes with various illumination, it is essential to utilize a flexible light source representation that adapts to different lighting and supports realistic rendering. In this work, we use a spherical harmonic illumination model to approximate the global illumination around a 3D object. Specifically, for each scene containing a car, we represent the lighting parameters using 3rd-order spherical harmonic coefficients, denoted as $\mathrm{L} \in \mathbb{R}^{16 \times 3}$, the incident radiance $\mathrm{L_i}$ in Eq. 3 is defined as:

$$\mathrm{L_i(x, \omega_i)} = \mathrm{L_i(\omega_i)} = \mathrm{L} \cdot \mathrm{SH}(\omega_\mathrm{i}), \tag{6}$$

where $\mathrm{SH}(\cdot)$ represents the spherical harmonic basis functions.

**Physically-based rendering**. Given the material attributes, the shading color for each Gaussian primitive can be computed. Since the definite integral in Eq. 3 is analytically intractable, we approximate it using numerical integration. For each Gaussian primitive G, we generate M incident ray directions through Fibonacci sphere sampling based on the normal direction. Since the scene contains only a single object and the geometry of a vehicle can be approximated as convex, we assume full visibility of the object to the environment lighting, setting the visibility factor for these ray directions to 1. Consequently, the physically-based shading color $\widetilde{\mathrm{c}}_\mathrm{i}$ of a Gaussian primitive $\mathrm{G_i}$ for outgoing direction $\omega_o$ can be formulated as:

$$\widetilde{\mathrm{c}}_\mathrm{i}(\omega_\mathrm{o}) = \frac{1}{\mathrm{M}} \sum_{\mathrm{j}=1}^{\mathrm{M}} \mathrm{f_r}(\mu_\mathrm{i}, \omega_\mathrm{j}, \omega_\mathrm{o}; \mathrm{n_i}, \mathrm{b_i}, \mathrm{r_i}, \mathrm{m_i}) \mathrm{L_i}(\omega_\mathrm{j}) \cos \theta, \tag{7}$$

The physically-based rendered image $\widetilde{\mathrm{C}}$ is then obtained using the differentiable tile-based rasterizer.

## 3.3 RELIGHTABLE 3D-GS GENERATIVE MODEL

Given a single input image I, we initially process it through a multi-view generation model Wonder3d Long et al. (2024) finetuned on Carverse to produce images from six predefined viewpoints, covering the front, back, left, right, left-front, and right-front perspectives of the vehicle. With these multi-view images $\mathrm{I_m}$ and corresponding camera poses $\pi_\mathrm{m}$ as input, we then exploit the interpolation capabilities of the transformer architecture and introduce an asymmetric U-Net Transformer $g_\theta$ as the Relightable 3D-GS Generative Model (RGM), to estimate the corresponding relightable GS and global illumination parameters. This can be mathematically expressed as:

$$\mathrm{G}, \mathrm{L} = g_\theta(\mathrm{F}), \tag{8}$$

$$\mathrm{C}, \widetilde{\mathrm{C}}, \mathrm{P} = Renderer(\mathrm{G}, \mathrm{L}, \pi), \tag{9}$$

Figure 3: **Overall architecture of RGM**. We first input a single image into the multi-view generation model to obtain multi-view consistent images of the car. These images and camera embeddings are then fed into the relightable 3D-GS generative model, which produces both global illumination parameters and relightable Gaussian representations. Through a physically-based rendering layer, we can obtain the material properties of the car and relight the car with a new illumination.

$G = \{G_0, G_1, ...G_K\}$ is a set of pixel-aligned relightable Gaussian primitives, L denotes the SH coefficients, and $F = \{f_0, f_1, f_2, ...\}$ is the embeddings of input images and camera poses. $Renderer(\cdot)$ is the physically-based rendering function and differentiable rasterizer, $P = \{N, B, R, M\}$ are the rendered material maps under camera pose $\pi$. Following LGM Tang et al. (2024a), we utilize the Plücker ray embedding to densely encode the camera poses. The RGB values and ray embeddings are concatenated into a 9-channel feature map, which serves as the input to RGM:

$$f_i = \{v_i, o_i \times d_i, d_i\}, \tag{10}$$

where $f_i$ is the feature for pixel i, $v_i$ is the RGB value, $d_i$ is the ray direction, and $o_i$ is the ray origin.

The architecture of $g_\theta$ is a hybrid model that integrates the hierarchical structure of U-Net for local feature extraction with self-attention mechanisms to capture global dependencies. It consists of three components: (1) Down Blocks to extract features from the input images while progressively reducing spatial resolution and increasing channel depth; (2) Mid Blocks to refine the features at the lowest resolution; and (3) Up Blocks to upsample the features to higher spatial resolutions, with skip connection that concatenate the upsampled features with corresponding features from the Down Blocks, allowing the network to retain fine-grained details. With the pixel-aligned features from the last U-Net layer, we then decode them through two prediction heads, producing spherical harmonics and relightable Gaussian parameters, respectively.

During training, we optimize RGM using a loss function that accounts for both rendered images and material properties. Denoting the ground-truth images as $\hat{C}$, the total loss is formulated as:

$$\mathcal{L}_{\text{total}} = \lambda_1 \|C - \hat{C}\|_2^2 + \lambda_2 \|\widetilde{C} - \hat{C}\|_2^2 + \lambda_3 \mathcal{L}_{\text{material}} + \lambda_4 \mathcal{L}_{\text{consistency}} + \lambda_5 \mathcal{L}_{\text{smooth}}, \tag{11}$$

where $\mathcal{L}_{\text{smooth}}$ is a bilateral smooth loss for materials and $\lambda_i$ is the loss weight for each term.

### 3.4 APPLICATIONS FOR AUTONOMOUS DRIVING SIMULATION

To demonstrate the practicality of RGM, we further present a 3D-GS-based pipeline for autonomous driving scene simulation that seamlessly integrates a vehicle into a dynamic road scene with photorealistic rendering. Given a set of posed images of the scene, we first employ a dynamic 3D-GS Chen et al. (2023b) to reconstruct the road scenes. Subsequently, we adopt DiffusionLight Phongthawee et al. (2024), a diffusion-based lighting estimation module, to generate the environment map from an image of the scene. Using an input image of the vehicle, RGM infers and generates the vehicle's Relightable GS, which is subsequently relighted using the environment map. The relighted GS is then spatially aligned to place the vehicle on the road, with its trajectory defined. Finally, the relighted GS is integrated into the road scene's dynamic 3D-GS, enabling the realistic rendering of vehicles within various road environments.

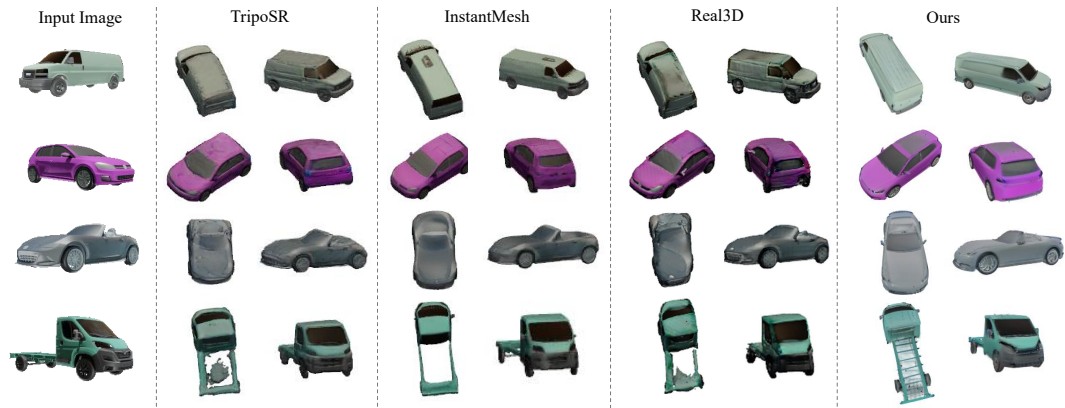

Figure 4: Qualitative results of generated novel views, compared with TripoSR Tochilkin et al. (2024),InstantMesh Xu et al. (2024) and Real3D Jiang et al. (2024).

| Method | PSNR ↑ | SSIM ↑ | LPIPS ↓ | Chamfer Dist. ↓ | F-score ↑ |
|---|---|---|---|---|---|
| LGM Tang et al. (2024a) | 14.95 | 0.843 | 0.312 | 0.171 | 0.183 |
| TripoSR Tochilkin et al. (2024) | 14.60 | 0.826 | 0.230 | 0.073 | 0.201 |
| InstantMesh Xu et al. (2024) | 14.27 | 0.827 | 0.243 | 0.038 | 0.404 |
| Real3D Jiang et al. (2024) | 14.57 | 0.823 | 0.224 | 0.079 | 0.165 |
| Ours | 20.16 | 0.871 | 0.094 | 0.031 | 0.512 |

Table 2: Comparison with state-of-the-art methods on Carverse test set.

## 4 EXPERIMENTS

### 4.1 COMPARISON WITH SOTA METHODS

We benchmark our approach on the Carverse dataset using 50 car samples for testing. Each test sample includes a single input image, along with a corresponding point cloud and 30 images from random views for evaluation. For the quantitative assessment of the generated images, we apply standard image quality metrics, including PSNR, SSIM, and LPIPS. To evaluate the accuracy of the reconstructed 3D shapes, we use Chamfer Distance (cm) and F-score (w/ threshold of 0.01m) as the evaluation metrics. Several state-of-the-art feed-forward image-to-3D methods Tochilkin et al. (2024); Xu et al. (2024); Tang et al. (2024a); Jiang et al. (2024) are selected for comparison. We register the generated meshes of these methods with ground truth mesh for rendered image evaluation. The quantitative results, presented in Tab. 2, show that our method significantly outperforms the others in generating car assets with accurate geometry and realistic textures from a single RGB image. Qualitative comparison examples are provided in Fig. 4, where our method demonstrates superior consistency and texture realism, largely due to the large-scale dataset that provides a robust foundation for generating high-quality 3D car assets. Moreover, RGM can generate the normal and material maps of cars as illustrated in Fig. 1, while other methods are limited to texture generation.

### 4.2 ABLATION STUDIES

**Supervision for Materials**. We also conduct an ablation study to evaluate the impact of material supervision by removing the loss term $\mathcal{L}_{\text{material}}$. The results, presented in Tab. 3, demonstrate a slight performance drop, but the model still achieves good results overall. This indicates that RGM can also be effectively trained on datasets lacking ground truth material information.

| Method | PSNR ↑ | SSIM ↑ | LPIPS ↓ | Chamfer Dist. ↓ | F-score ↑ |
|---|---|---|---|---|---|
| w/o $\mathcal{L}_{\text{material}}$ | 19.85 | 0.866 | 0.123 | 0.054 | 0.352 |
| Ours | 20.16 | 0.871 | 0.094 | 0.031 | 0.512 |

Table 3: Ablation studies on material supervision.

**Relightable Gaussian**. To investigate the effect of the relightable Gaussian, we perform an ablation study by replacing the relightable Gaussian with the vanilla 3D-GS, training with the same dataset. Fig. 5 provides a visualization of generated Gaussian points training with and without relightable Gaussians. Removing the relightable Gaussian results in noisy Gaussian points around the car, while the relightable Gaussian points are clean and well-defined. We credit this enhancement to the normal attributes, which offer a deeper understanding of the 3D objects and facilitate the geometry learning of the 3D-GS.

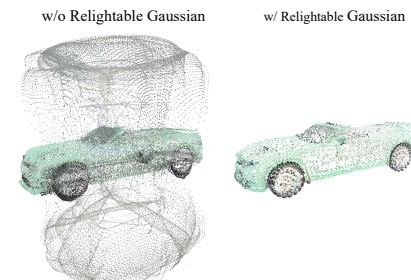

Figure 5: Visualization of gaussian points with or without relightable gaussians.

### 4.3 APPLICATIONS

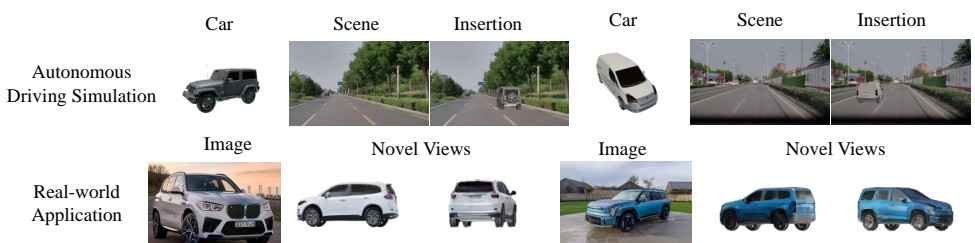

Figure 6: Qualitative results for applications.

**Applications for Autonomous Driving Simulation**. We present the results of the autonomous driving simulation detailed in Sec. 3.4. As illustrated in Fig. 6, the system successfully achieves photorealistic rendering, seamlessly inserting the given vehicle into the road scene and the appearance of the car matches the lighting naturally.

**Real-world Applications**. To further validate the effectiveness of RGM, we also applied it to in-the-wild images of real-world cars. The qualitative novel view images are given in Fig. 6. This shows the capability of RGM to transform a random car on the road into a 3D digital asset.

## 5 CONCLUSION

In conclusion, in this paper we successfully tackle the challenges in 3D vehicle generation, such as limited datasets and lack of modeling of material properties for relighting. By introducing the Carverse dataset, containing over 1,000 high-precision 3D vehicle models, we support robust learning and generalization. Additionally, our novel relightable 3D-GS generation framework named RGM enables the efficient creation of detailed 3D car models by modeling global illumination and representing 3D scenes through Gaussian primitives that capture geometry, appearance, and material properties. This approach allows for realistic relighting under different illumination conditions, advancing 3D asset generation for AR/VR applications and industrial design by improving both speed and quality in automatic 3D car modeling from a single image.

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
