# OpenReview forum: "RGM: Reconstructing High-fidelity 3D Car Assets with Relightable 3D-GS Generative Model from a Single Image"
_ICLR.cc/2025/Conference — ICLR 2025 Conference Withdrawn Submission_

### Official Review · Reviewer_2d22 · 2024-10-28

**Soundness:** 3
**Presentation:** 3
**Contribution:** 2
**Rating:** 3
**Confidence:** 4

**Summary:**

The paper addresses the task of relightable object generation, overcoming the previous technical challenge of generating only Lambertian objects under fixed lighting conditions. The key insight and motivation behind the paper's approach is the construction of a large-scale synthetic car dataset, the representation of 3D objects using global illumination and relightable 3D Gaussian primitives, and the use of a feed-forward model to directly predict relightable Gaussian primitives and global illumination. The specific methodology involves taking a single image as input and employing a multi-view diffusion model to generate multi-view images. These multi-view images, along with their corresponding camera ray embeddings, are then fed into the feed-forward model to predict the relightable Gaussians and global illumination, enabling the generation of objects that can be relit under various lighting conditions.

**Strengths:**

1. The large-scale car dataset is a good contribution.
2. The paper is well organized and easy to read.

**Weaknesses:**

1. The insertion effect in Figure 6 of the paper is not satisfactory; the reconstructed car appears artificial when placed into the scene. The authors need to explain the reasons behind this. I suspect it is due to the lack of reflective texture on the car and the absence of shadow modeling. While I can accept the lack of shadow modeling, the absence of reflective texture implies inaccurate material prediction. Is this due to the poor quality of the dataset created in the paper?
2. The core contribution of the paper is the prediction of relightable objects, but the experiments lack validation of relighting, including the accuracy of materials and relighting. The paper also does not compare with methods that do relightable object generation, such as UniDream. Without these experiments, it is impossible to see the effectiveness of the paper's method. Moreover, the paper lacks relevant visualization results.
3. The paper states that the feed-forward model outputs pixel-aligned Gaussian primitives. Does this refer to Gaussian maps? This is not clear. If it is Gaussian maps, Figure 3 only shows the Gaussian maps from a single perspective, and the paper does not mention whether it is multi-view. If it is only a single perspective, how can it represent a 360-degree object?
4. In Figure 3, the feed-forward model is referred to as the "Relightable 3D-GS Generative Model", but based on the text description of the paper, I do not see that this GS prediction model is a generative model. Why is it called a generative model in Figure 3? Or is it just a typo?
5. The technical contribution of the paper is relatively weak, mainly adding the prediction of relightable Gaussian primitives to LGM. I believe this is not sufficient for the ICLR conference.

**Questions:**

The result quality and the limited contribution should be clearly addressed.

---

### Official Review · Reviewer_tR3d · 2024-10-28

**Soundness:** 3
**Presentation:** 3
**Contribution:** 2
**Rating:** 6
**Confidence:** 4

**Summary:**

This paper presents a novel approach for generating high-quality 3D car assets with material properties from a single image, demonstrating promising results. However, there are a few aspects that could be improved for further advancement.

**Strengths:**

This paper presents a novel approach for generating high-quality 3D car assets with material properties from a single image, demonstrating promising results.

**Weaknesses:**

Dataset Generalization: The method heavily relies on the Carverse dataset, which may limit its applicability to other types of 3D objects. Exploring data augmentation techniques or constructing additional datasets for different object categories would enhance the model’s generalization capabilities.
Model Efficiency: The current model, based on U-Net Transformer, incurs high training costs. Investigating more lightweight architectures or efficient training strategies could improve the model’s practicality.
Detail Generation: While the method achieves impressive results for overall car structures, the generation of fine details remains limited. Incorporating techniques to better capture intricate features would be beneficial.
Real-time Rendering: The differentiable rasterizer employed in the model compromises real-time performance. Exploring more efficient rendering methods would enable the model to be utilized in real-time applications.
Model Interpretability: The black-box nature of the model hinders understanding its inner workings. Investigating methods to enhance model interpretability would provide valuable insights into the generation process and potential limitations.

**Questions:**

How to obtain the corresponding camera poses of the multi-view images?
Does this work need the camera pose of the input image and how to obtain it?

---

### Official Review · Reviewer_KWT2 · 2024-11-01

**Soundness:** 3
**Presentation:** 3
**Contribution:** 2
**Rating:** 5
**Confidence:** 4

**Summary:**

This paper introduces RGM (Relightable 3D-GS Generative Model), a framework designed to generate high-fidelity, relightable 3D car models from a single image. The study includes the development of Carverse, a synthetic dataset with over 1,000 3D car models, providing robust training data. The RGM framework combines 3D Gaussian splatting with physically-based rendering, enabling realistic, material-consistent models adaptable to various lighting scenarios. Experimental results show that RGM surpasses previous methods in terms of image quality, geometric accuracy, and material consistency, with potential applications in autonomous driving simulation.

**Strengths:**

1. Clear Writing: The paper is well-structured and clearly written, with logical flow and coherent narrative, making the motivation, methodology, and experiments easy to follow.
2. Contribution of Dataset Construction: The authors constructed a large-scale, high-quality synthetic dataset, Carverse, with over 1,000 3D vehicle models and extensive multi-view and material attribute data. This dataset serves as a valuable foundation for 3D car model generation and advances the field by supporting further research.

**Weaknesses:**

1. Lack of Real Data Testing: The experiments were conducted mostly on synthetic data, the paper only presents two qualitative results on real data in section 4.3. The authors could conduct more experiments on real data, especially providing some quantitative results.
2. Insufficient Analysis on Viewpoint Consistency: The paper lacks an analysis of material consistency between generated multi-view images and the input image. Without ensuring consistency in material properties across views, the relightable 3D-GS model may have limited relevance, potentially restricting its applicability.
3. Moderate Novelty: The design of the relightable 3D Gaussian splatting model and the generative model shows limited innovation compared to existing methods. While performance improvements are achieved, the methodological design lacks significant originality.

**Questions:**

1. The results of both the generated novel view images and the final 3D-GS rendered results are of relatively low resolution. It is recommended that the authors could provide some high-resolution results or zoomed-in images of specific areas to better illustrate the effects in detailed regions.
2. It is unclear whether the compared methods were trained on the same dataset. The authors could clarify the training data used for each compared method in their experimental setup, and if different, discuss how this might impact the fairness of the comparisons.

---

### Official Review · Reviewer_8zJ4 · 2024-11-04

**Soundness:** 1
**Presentation:** 2
**Contribution:** 3
**Rating:** 3
**Confidence:** 3

**Summary:**

The main contribution of the paper is the introduction of Relightable 3D-GS Generative Model (RGM), a novel framework for automatically generating high-fidelity 3D car assets from a single image. RGM addresses the limitations of existing models by enabling relighting under various lighting conditions. It achieves this through the use of 3D Gaussian primitives integrated with BRDF material properties and global illumination modeling. Additionally, the authors introduce the Carverse dataset, a large-scale synthetic collection of over 1,000 detailed 3D vehicle models, to train and benchmark their approach.

**Strengths:**

The relightable 3DGS car model is interesting and worth exploring, The RGM method enables the creation of high-quality 3D car models from a single image, with realistic relighting features, addressing key shortcomings in existing 3D generation techniques.

The Carverse dataset offers a large-scale collection of 3D vehicle models, and what sets it apart is the inclusion of detailed material and normal information. This gives it a unique edge compared to older datasets, making it useful for robust training and improving the realism of generated 3D assets.

If the relighting capability works as well as claimed, this paper shows strong potential for practical applications, especially in fields like autonomous driving,  where realistic lighting is crucial.

**Weaknesses:**

For table 1, the use of terms like 'high' and 'low' for the quality and diversity metrics is very ambiguous and biased, especially when many of the datasets being compared are real-world datasets. From the images presented in the paper, I don't observe a significant quality improvement solely based on these visuals. However, I do recognize the added value that Carverse provides through its inclusion of normal and material information, which may not be available in the other datasets mentioned. That said, when comparing diversity and quality, datasets such as CarStudio visually appear much more diverse and of higher quality to me. Therefore, using 'high' to describe both Carverse and CarStudio can mislead readers into thinking their visual appearances in terms of diversity and quality is comparable, which doesn’t seem accurate. If we further consider the difference in instance numbers, CarStudio’s over 200k instances are much denser than the 1,000 instances introduced by Carverse.

The experimental section appears biased. If my understanding is correct, when the authors compared the RGM model’s ability to generate car models with other SOTA models, they used 50 car samples from their own Carverse dataset. However, only the RGM model was trained on the Carverse dataset, while the other models had no prior exposure to this dataset during training. This creates an inherent advantage for the RGM model, as it has a better understanding of the Carverse data distribution, which would inevitably lead to better quality metric scores compared to the other models.

The authors claim that the quality of existing synthetic car datasets is insufficient for modern real car rendering, but from visual inspection, I don’t see a significant quality improvement in Carverse compared to other synthetic datasets like Objaverse. In fact, in some instances, Objaverse’s car models even appear to have better visual quality. To truly demonstrate that Carverse is superior to previous datasets, a more convincing A/B test with identical conditions under RGM would be necessary.

I would also like to see more examples of scene insertion results to better assess how well the RGM model handles relighting. In Figure 6, both scenes shown are under bright daylight conditions. It would be more convincing to see examples in dimmer lighting environments to showcase how the relighting function performs under different lighting scenarios.

**Questions:**

Q1. You mention that RGM supports relighting through spherical harmonics. How flexible is the lighting adjustment when integrating the 3D models into new scenarios? For example a scene with much dimmer environmental lighting. So far I only saw bright daylight insertion in the paper.

Q2. How does RGM handle complex lighting environments like dynamic or moving light sources?

Q3. In Table 1, terms like "high" and "low" are used to describe dataset quality and diversity. Can you clarify how these metrics are quantitatively defined? What specific attributes (maybe resolution, texture detail, polygon density) are used to distinguish between "high" and "low"?

Q4. When conducting comparison experiments with other SOTA models, were those models trained or fine-tuned on Carverse before testing? If not, and only RGM has prior knowledge of the distribution of the Carverse data, how do you address the potential bias in comparing RGM's performance to SOTA methods that were not trained on the Carverse dataset?

---

### Note · Authors · 2024-11-15

I have read and agree with the venue's withdrawal policy on behalf of myself and my co-authors.